# Exploratory study on the impact of *Ganoderma australe* extract on gut microbiota and immune gene expression in honey bees exposed to *Vairimorpha ceranae*

**Sarah Zuern[1]\*, Bella Romero[1], Carlos Spichiger[1], Leandro Ortiz[2], Alejandro Jerez[3], Esteban Basoalto[4], Max Emil Schön[5]\*, Sigisfredo Garnica[1]\***

**1** Universidad Austral de Chile, Instituto de Bioquímica y Microbiología, Valdivia, Chile, **2** Universidad Austral de Chile, Instituto de Química, Valdivia, Chile, **3** Universidad Austral de Chile, Instituto de Farmacia, Valdivia, Chile, **4** Universidad Austral de Chile, Instituto de Producción y Sanidad Vegetal, Valdivia, Chile, **5** Department of Biomolecular Mechanisms, Max Planck Institute for Medical Research, Heidelberg, Germany

\* sigisfredo.garnica@uach.cl (SG); max-emil.schoen@mr.mpg.de (MES); sarah.zuern@uach.cl (SZ)

## Abstract

The microsporidium *Vairimorpha* (*Nosema*) *ceranae* is an emerging threat to honey bees (*Apis mellifera*), known to disrupt gut microbiota and suppress immune responses, potentially contributing to colony losses. Fungal extracts have recently gained interest as sources of bioactive compounds with antimicrobial and immunomodulatory potential. In this study, we explored the effects of different dietary supplements—sugar syrup, HiveAlive™, and a novel *Ganoderma australe* extract (GanoBee)—on gut bacterial composition and immune-related gene expression in honey bees subjected to experimental exposure to *V. ceranae* $1 \times 10^4$ spores per bee. The GanoBee diet altered the gut microbiota, notably reducing the relative abundance of Rhizobiaceae (*Bartonella apis*) and increasing *Frischella* compared to other treatments. While alpha diversity was not significantly affected by diet or exposure to *V. ceranae*, beta diversity differed significantly in bees fed with GanoBee. Additionally, the expression of the antimicrobial peptide genes *abaecin* and *hymenoptaecin* was elevated in both exposed and unexposed bees fed with GanoBee, depending on the sampling day. However, the establishment of *V. ceranae* infection appeared limited, likely due to low spore viability, and mortality in control bees was higher than expected. The low *Vairimorpha ceranae* infection levels observed in this study are likely attributable to reduced spore viability caused by storage conditions and/or suboptimal environmental conditions within the laboratory cages. Post hoc analyses indicated that the high viscosity of GanoBee-supplemented diets likely contributed to the elevated bee mortality observed, underscoring a critical limitation of the experimental design related to diet formulation and delivery method. These physical factors complicate the interpretation of treatment efficacy and highlight the importance of

**Data availability statement:** Raw sequencing data files are available in the NCBI Sequence Read Archive (SRA) under BioProject accession number PRJNA1347411(http://www.ncbi.nlm.nih.gov/bioproject/1347411). The bioinformatics workflow (flowchart) is available in the GitHub repository "16S-Assignments-DADA2" (https://github.com/Pcariman/16S-Assignments-DADA2). All relevant data are within the manuscript and its Supporting Information files.

**Funding:** Funding for this project was made possible by project the Applied Research and Innovation 2023 grant INID210009. ME. Schön would like to thank I. Schlichting and the Max Planck Society for their support. The funders had no role in study design, data collection and analysis, decision to publish, or preparation of the manuscript.

**Competing interests:** The authors have declared that no competing interests exist.

optimizing feeding protocols to avoid confounding effects. Despite these constraints, GanoBee demonstrated promising potential as a modulator of gut microbiota composition and immune-related gene expression, supporting the need for further research under improved and carefully controlled experimental conditions.

## Introduction

Honey bees serve as essential pollinators, playing a vital role in supporting agricultural productivity and preserving biodiversity across ecosystems [1,2]. The western honey bee (*Apis mellifera* L.) is globally distributed and ranks as the most frequent pollinator species for crops worldwide, contributing to food production and biodiversity [3]. Over the last decade, a global decline in pollinators has been observed, marked by a reduction in wild pollinator populations and alarming losses of honey bee species [4,5]. One of the contributing factors of the honey bees decline include pathogens such as *Vairimorpha* (*Nosema*) *ceranae*, an intracellular microsporidian first described in the Asian honey bee (*Apis cerana*) [6] which causes the infectious disease nosemosis in honey bees, predominating over *V. apis* in Chile and other countries [7]. This pathogen degenerates gut epithelial cells, leading to digestive problems and diarrhea which impair the honey bee's ability to absorb nutrients properly [8]. Nosemosis results in shorter lifespan, changes in metabolism and immune suppression in honey bees [9]. The constant release of infectious spores results in a rapid spread of the disease in the hive and can even lead to colony collapse [10].

Several studies have investigated the immune response of honey bees to *V. ceranae* infection but the results are inconsistent [11]. According to Antúnez et al. [12], the transcript levels of antimicrobial peptides (AMPs) and vitellogenin in *V. ceranae*-inoculated honey bees were significantly downregulated compared to the control group. Further studies confirmed the downregulation of immune-related gene expression in *V. ceranae*-infected honey bees [13–15]. These results indicate that *V. ceranae* infection leads to immunosuppression in honey bees. However, other studies have reported a significant upregulation in the expression of immune-related genes in honey bees infected with *V. ceranae* [16,17].

Honey bees emerge from the pupal stage without their core gut microbiota and acquire it within four to six days through social interactions, food consumption and contact with hive surfaces [18]. A stable gut microbiota is essential for digestion, detoxification of food components, metabolism, modulation of the immune system and protection against pathogens [19]. According to previously published results, the diet of honey bees [20–23] and *V. ceranae* infection [24–27] alter the composition and diversity of the gut microbiota.

Over the past decade, *V. ceranae* has emerged as a globally prevalent pathogen in honey bees posing a growing challenge for beekeepers worldwide [7]. The only effective chemical treatment against nosemosis in apiculture is the antibiotic fumagillin [28]. However, the use of fumagillin in apiculture may pose a risk to human health due to potential residues in honey and has been reported to negatively affect bee health [29].

There are numerous plant-based dietary supplements for honey bees available on the market, such as, ApiHerb, Honey-B-Healthy®, Nozevit and HiveAlive™ [9,30]. The efficacy of these products against *V. ceranae* has been tested, with HiveAlive™ showing the most promising results. Thus, during a two-year trial, HiveAlive™ reduced the spore load of *V. ceranae* and significantly increased hive population compared to the control group [31]. Garrido et al. [32] confirmed the effectiveness of HiveAlive™ in reducing the spore load of *V. ceranae*. However, in a field experiment the treatment with HiveAlive™ increased the spore intensity of *Vairimorpha* spp. in honey bees [33].

Recent research highlights the potential of fungal extracts as an option to enhance bee health [34]. According to Glavinic et al. [35,36], fungal extracts reduced *V. ceranae* spore load, improved survival rates and upregulated gene expression of AMPs and vitellogenin in *V. ceranae*-infected honey bees. In addition, polysaccharide extracts derived from the mycelia or fruiting bodies of *Ganoderma australe* demonstrated various biological activities, including immunomodulatory, anticancer, antioxidant, and antimicrobial activities [37–41]. Despite the recognized beneficial properties of *G. australe*, its extracts have never been explored on honey bees.

The aim of this study was to evaluate the potential effects of a novel *G. australe* extract (GanoBee) on gut bacterial microbiota and immune-related gene expression in honey bees subjected to experimental *V. ceranae* exposure. We hypothesized that dietary supplementation with GanoBee would influence gut microbial composition and immune responses, potentially providing health benefits comparable to or greater than those of the commercial product HiveAlive™. To test this, newly emerged bees were assigned to different dietary treatments—sugar syrup, HiveAlive™, or three concentrations of GanoBee—and half of the groups were exposed to *V. ceranae* on day 7 post-emergence.

Gut bacterial communities were assessed by 16S rRNA amplicon sequencing to evaluate diet- and exposure-related shifts in microbiota. Immune response was examined by RT-qPCR analysis of selected AMP genes and vitellogenin. Additionally, we explored potential correlations between gut microbiota composition and immune gene expression. However, we acknowledge that the *V. ceranae* infection may not have been fully established, and the observed control mortality was higher than expected. Therefore, the findings presented here should be considered preliminary and interpreted with caution, serving as a basis for future research under improved experimental conditions.

## Materials and methods

### Ethics statement

The European honey bee (*Apis mellifera*) is neither an endangered nor protected species. No specific permits were required for the described study. The apiary is part of Universidad Austral de Chile's facilities and is not privately-owned or protected.

### Fruiting bodies collection and pure cultures

Fruiting bodies of *Ganoderma australe* (Fr.) Pat. were collected from a living *Nothofagus obliqua* (Mirb.) Oerst tree in the Fundo Teja Norte (39°47'54.01"S, 73°16'0.68"W) located in the commune of Valdivia belonging to Los Ríos region in Chile (S1 Fig). Fungal cultures were obtained from a young and healthy fruiting body, and isolations were carried out aseptically in the Mycology Laboratory of the Universidad Austral de Chile. Small portions were placed in Petri dishes containing 2% malt extract agar (MEA) with two to three drops of previously sterilized antibiotic solution containing Penicillin G 0.4 g and Streptomycin 9 mg in 200 ml, respectively. In our study, cultures were maintained at 23°C and 60% humidity for a maximum of 30 days.

To check for mycelial growth and possible contamination, cultures were examined every two days. The cultures were analyzed after 14 days using a light microscope by comparing the microscopical structures with previously published culture descriptions [42]. The isolated strain as well as the corresponding fruiting body and substrate are deposited in the fungal collection of the Mycology Laboratory of the Universidad Austral de Chile.

## Molecular identification of fungal isolates

For molecular identification, total DNA was extracted from each strain exhibiting the morphological characteristics of *Ganoderma* described by Stalpers [42] using the E.Z.N.A. Fungal DNA Mini Kit (Omega Bio-Tek Inc). Small pieces of agar mycelium were put into a 1.5 ml centrifuge tube. The fungal material was macerated using a sterile pistil and 300 µl FG1 buffer was added. Subsequently, the total volume of the homogenate was transferred into a 2.0 ml microtube containing silica lysis beads (0.5 mm) to destroy the hyphae for two periods of 5 min each in a Mini-BeadBeater-16 Model670EUR (BioSpec Products, Inc., USA) at 253×g. The following DNA extraction steps were performed according to the manufacturer's instructions.

To amplify the nuclear segment including the internal transcribed spacer (ITS) region (ITS1-5.8S-ITS2), the combination of the ITS1-F primer 5'-CTTGGTCATTTAGAGGAAGTAA-3' [43] and the ITS4 primer 5'-TCCTCCGCTTATTG ATATGC-3' [44] was used. A touchdown PCR protocol was applied with an initial step of 94°C for 3 min, followed by 10 cycles of 94°C for 30 s, with annealing temperatures starting at 60°C for 45 s (decreasing by 1°C/cycle), then 72°C for 1 min 15 s for elongation, followed by 26 cycles of 94°C for 30 s, 50°C for 45 s, elongation at 72°C for 1 min 15 s and a final extension at 72°C for 7 min. The same primers were used for PCR and sequencing of both DNA strands on an automated genetic analyser at the Austral-omics center of the Universidad Austral de Chile. The Geneious Prime software v2022.2 was used to manually edit chromatograms of forward and reverse sequences, generate contigs and consensus sequences. The consensus sequences were submitted to BLAST (Basic Local Alignment Search Tool search) for identification of the fungal isolates.

## Fungal extract

Once purity and taxonomic identification had been confirmed, the strain was grown on a 2% malt extract agar medium. Fungal culture was grown for three weeks in laboratory conditions. Finally, the powdered mycelia of *G. australe* were added to the preserved syrup to obtain a 0.5%, 1.0%, and 1.5% extract solution (GanoBee) (S2 Fig).

Polysaccharides of *G. australe* mycelia were extracted using a technique modified from Oberemko et al. [45] and Câmara et al. [46] and analyzed by Fourier transform infrared (FTIR) spectroscopy [47] and hydrolysis with Molish's reagent [48] in the Laboratory of Natural Products belonging to the Institute of Chemical Sciences. Additional chemical compounds were analyzed in the Laboratory of Phytotechnology. Ganoderic acids were detected with a high-performance liquid chromatography (HPLC) analysis using an Agilent 1200 Series system equipped with a diode array detector (MWD 61365 D, Agilent). A ganoderic acid A standard (CAS No. 81907-62-2) was used at a concentration of 50 µg/mL and dissolved in 100% acetonitrile. The standard was purchased from Sigma-Aldrich (Chile). The analytical procedure was modified from Liu et al. [49]. The detection wavelength was set at 250 nm. Separation was achieved on an Agilent Zorbax 300SB column (5 µm, 4.6 mm×250 mm), maintained at 30°C. The mobile phase consisted of a gradient elution using (A) acetonitrile and (B) 0.1% acetic acid in water (v/v). The injection volume was 20 µL, and the flow rate was set to 1.0 mL/min in continuous mode. Data acquisition and analysis were conducted using Agilent ChemStation Software.

## Experimental design and honey bee sampling

Honey bees were obtained from the experimental apiary belonging to the Faculty of Agronomy and Food Science of the Universidad Austral de Chile. The hives form part of the infrastructure used for undergraduate teaching, and a continuous record of colony health has been maintained through routine visual inspections. These evaluations include assessments of queen presence, overall colony behavior, brood pattern, foraging activity, occurrence of ectoparasites and endoparasites (as *Varroa destructor* or *Acarapis woodi*), and visible symptoms of disease. For the present study, additional diagnostic analyses were performed, including microscopic examination of bee samples to detect *Vairimorpha* spp. spores using standard procedures [50]. Only hives that were vigorous and showed no symptoms of any disease were used in the

experiments. Wooden cages (12 x 12 x 4.5 cm) were especially designed for this experiment, and a fine wire mesh has been attached to the front of the cages to allow honey bees to be observed (S2 Fig). The removable sliding side at the back of the cage made it possible to remove dead honey bees and clean the cages. Each cage contained a water feeder, a 2.0 ml microcentrifuge tube modified by drilling four holes (Ø 2.0 mm) in a line [51] Honey bees were fed using a gravity feeder constructed by drilling eight Ø 2.0 mm diameter holes in the base of an inverted 15 mL Falcon conical tube. Feeding tubes were replaced daily with a fresh diet solution.

Frames with sealed brood were taken from five chosen colonies and incubated at 30°C ± 1°C in the dark in the Laboratory. Newly emerged honey bees were transferred to 30 experimental cages (3 cages per treatment). Each cage contained approximately 50 honey bees. Ten groups were established, and the following treatments were performed from the first day after emergence until the end of the experiment (Table 1).

Honey bees were divided into ten different treatment groups (3 cages/treatment). From the beginning of emergence, honey bees were fed with different diets until the end of the experiment. On the 7th day post-emergence five treatment groups were infected with *V. ceranae*. Five honey bees were collected from each cage on different sampling days and stored at −80°C.

The control groups (C and CN) received 50% sucrose solution (w/v). Honey bees of the HA and HAN group were treated with 2.5 ml of HiveAlive™ per liter of 50% sucrose solution (w/v), a widely used commercial plant-based supplement that serves as a positive control. The other six groups were treated with different concentrations (w/v) of GanoBee: 0.5% (GB-1 and GBN-1); 1.0% (GB-2 and GBN-2) and 1.5% (GB-3 and GBN-3). Honey bees were fed *ad libitum* with their corresponding solution. On the 7th day post-emergence five treatment groups were infected with *V. ceranae* (details see below). Honey bees were incubated at 29°C, dead honey bees were removed daily, and their numbers were recorded. For gene expression, spore counting and microbiota analysis, five honey bees were collected from each cage on days 0, 3, 6, 9 and 12 after *V. ceranae* infection. Honey bees were transferred into a 15 ml centrifuge tube and put immediately in liquid nitrogen. Samples were stored at −80°C for further analyses.

## Microsporidia inoculum preparation and experimental infection of honey bees

Frozen honey bees naturally infected with *V. ceranae* were obtained from the APICOOP Ltda. cooperative and used for inoculum preparation. Several *V. ceranae*-infected honey bees were used for inoculation, which had been stored at −20°C for several days. The inoculum was prepared according to the protocol of Fries et al. [50]. Ten microliters of the spore suspension were loaded onto a hemocytometer and spores were quantified under a light microscope using the method of Cantwell [52]. Finally, the purified spore solution was diluted to a final concentration solution of $1 \times 10^6$ spores/ml with the corresponding treatment. Seven days after emergence, honey bees from the treatment groups CN, HAN, GBN-1, GBN-2 and GBN-3 were infected with *V. ceranae* spore solution. A total of five honey bees of each cage were fed individually 10 µl of *V. ceranae* spore solution using a micropipette according to the protocol described by Williams et al. [54]. Individual

**Table 1. Experimental design.**

| Treatment | Group | Diets after emergence | Sampling days (post-infection) |
|---|---|---|---|
| Non-infected | C | Sugar syrup (50% w/v) | 0 day |
| Non-infected | HA | HiveAlive™ | 3rd day |
| Non-infected | GB-1 | 0.5% GanoBee | 6th day |
| Non-infected | GB-2 | 1.0% GanoBee | 9th day |
| Non-infected | GB-3 | 1.5% GanoBee | 12th day |
| Infected | CN | Sugar syrup (50% w/v) | 3rd day |
| Infected | HAN | HiveAlive™ | 6th day |
| Infected | GBN-1 | 0.5% GanoBee | 9th day |
| Infected | GBN-2 | 1.0% GanoBee | 12th day |

fed honey bees were marked with a pen to distinguish them from the other honey bees. Additionally, honey bees were fed by group feeding, each with 2 mL of the spore solution prepared at the same concentration as previously described. Once the total infection dose was consumed, honey bees were fed *ad libitum* with their corresponding diet.

### Post hoc physical characterization of dietary supplements

Following unexpectedly high mortality rates observed during the experiment—particularly among bees fed with the GanoBee extract—we performed post hoc analyses of the physical properties of the administered diets. Specifically, we measured the density and viscosity of sugar syrup alone, sugar syrup supplemented with HiveAlive™, and sugar syrup supplemented with GanoBee at three concentrations. The GanoBee-enriched solutions exhibited significantly higher viscosity compared to the other diets. The density of sugar syrup, HiveAlive™, and three different concentrations of GanoBee (0.5%, 1.0% and 1.5%) were analyzed in the Laboratory of Phytotechnology. In addition, the viscosity of the different dietary supplements was determined in the Laboratory of Natural Products belonging to the Institute of Chemical Sciences. All analyses were performed in triplicate at a constant temperature of 30°C.

### Microscopic and PCR-based assessment of *Vairimorpha ceranae* infection

To analyze the *V. ceranae* infection on the different sampling days, the abdomens of five honey bees from each *V. ceranae*-infected treatment group (CN, HAN, GBN-1, GBN-2 and GBN-3) were separated and homogenized in 1 ml distilled $H_2O$. Ten microliters of the homogenate were loaded onto a hemocytometer, and the spore count was determined under a light microscope at a magnification of 400x according to the methodology described by Cantwell [52]. In addition, the presence of *V. ceranae* was confirmed by PCR analysis using specific primer for *V. ceranae*; Nc841f: 5'-GAGAGAA CGGTTTTTTGTTTGAGA-3' and Nc980r: 5'-ATCCTTTCCTTCCTACACTGATTG-3' [53].

### cDNA synthesis

For total DNA and RNA co-extraction, the Quick-DNA/RNA™ MiniPrep Plus Kit (Zymo Research) was used. First, the head of each bee was removed, and the abdomen was placed in a 2.0 ml microtube with 800 µl DNA/RNA Shield™. For homogenization, tungsten carbide beads (3 mm) were used, and samples were homogenized in a Mini-BeadBeater-16 at 253 × g for 1:20 min. The following steps were performed according to the manufacturer's instructions. To confirm that inoculation with *V. ceranae* was successful, PCR was performed as described previously. In addition, it was verified that honey bees from the non-infected treatment groups were *V. ceranae* free.

### Quantitative real-time PCR

The genes abaecin, hymenoptaecin and vitellogenin were quantified in this work. Since GAPDH and β-actin showed a consistent gene expression in various honey bee tissue samples [55], both genes were used as reference genes for target normalization in this study. Primer information is listed in S1 Table. Reverse transcription quantitative PCR (RT-qPCR) was performed using Takyon™ Rox SYBR® MasterMix dTTP Blue on a QuantStudio 3 Real-Time PCR System. The amplification was performed according to the following protocol: 50°C for 2 min, 95°C for 3 min, then 40 cycles of 95°C for 10 s and primer-specific annealing for 45 s. Specificity of each reaction was checked by a melt-curve analysis using the following protocol: 95°C for 15 s, 60°C for 1 min (rising 1.6°C/s), and 95°C for 15 s (0.15°C/s). To determine the optimal primer concentration and annealing temperature, the efficiency of each primer pair was analyzed. In each run non-template controls (nuclease-free water) were included, and samples were performed in duplicates. The amplification of each gene was expressed as a cycle threshold (Ct) value. The relative quantification of gene expression was calculated using the $2^{-\Delta\Delta CT}$ method [56]. It was ensured that the expression of the reference genes was consistent across all treatment groups. The geometric mean Ct of GAPDH and β-actin was used to normalize the expression of the target genes.

## 16S rRNA amplicon sequencing

Total DNA from all samples (n = 133) was quantified with the Qubit™ 4 Fluorometer, using the Qubit dsDNA HS Assay Kit. 30 pools were prepared by mixing an equal number of nanograms (ng) of the corresponding samples. To amplify the V3–V4 hypervariable region of the bacterial 16S rRNA gene, a PCR reaction was performed using the primers 341F: 5'-CCTACGGGNGGCWGCAG-3' and 785R: 5'-GACTACHVGGGTATCTAATCC-3' [57]. The library was constructed through a second PCR to add the Illumina P5 and P7 adapters with the Illumina Nextera XT Index Kit v2. The amplified libraries were purified with magnetic beads, eluted in 22 µl 10 mM Tris, and their integrity was assessed by electrophoresis. Subsequent size verification was performed using the Fragment Analyzer capillary electrophoresis system. The libraries were quantified by fluorometry and diluted to the appropriate loading concentration required on the NextSeq1000 sequencing system. The barcoded DNA amplicons were sequenced onto the Illumina sequencing platform using MiSeq Reagent Kit v3 (600-cycles), with FastQ configuration and paired-end sequencing. Amplicon sequencing of the gut bacterial microbiota and demultiplexing process of the raw reads were performed at the Austral-omics center of the Universidad Austral de Chile.

## Bioinformatic and statistical analyses

The quality metrics of the FASTQ files were assessed using the FastQC program. Adapter sequences were removed in the Linux terminal using Python with the Cutadapt package v3.7 and sequences were filtered with the Trimmomatic package v0.39. Sequences were filtered a second time based on sequence quality criteria observed in the quality plots of the DADA2 package [58] v1.16 from the BiocManager package v1.30.25 in R v4.3.1. Subsequently, the paired sequences (forward and reverse) were assembled and all sequences with a length below 134 bp were removed. Next, potentially chimeric sequences were removed, and the database was used for taxonomic annotation. Taxonomic assignment of each amplicon sequence variant (ASV) was performed by comparing sequences to the SILVA 138.1 reference database [59]. Finally, a phyloseq object was generated with the phyloseq package v1.50.0. and subsequent analyses were performed using R software v4.4.2. The relative abundance of each ASV was calculated and ASVs were grouped by treatment and sampling day at the family and genus level. All ASVs with a relative abundance of 0 were eliminated, and those with an abundance of less than 0.5% were categorized as 'Others'. The ASVs without taxonomic assignment but belonging to the Rhizobiaceae family were assigned at species level by comparing against references sequences from GenBank database. Alpha diversity was evaluated with the Shannon and Simpson diversity indices using the *estimate_richness* function, and significant differences were analyzed with Kruskal-Wallis test. Beta diversity among treatment groups was assessed at the genus level using Bray-Curtis dissimilarity and Binary Jaccard index (based on presence/absence data) and visualized through non-metric multidimensional scaling (NMDS). Statistical significance was determined with PERMANOVA (permutational multivariate analysis of variance), performed with the *adonis* function from the vegan package [60] v2.6-8, with a maximum of 999 permutations. To examine differences in dispersion among treatment groups, the *betadisper* function from the vegan package was used. Correlation analyses were conducted between the relative abundance of gut bacteria at the genus level and the gene expression of abaecin, hymenoptaecin and vitellogenin using the non-parametric Spearman rank test. The p-value was adjusted with a Bonferroni correction to account for multiple comparisons.

The survival probability of honey bees was analyzed using the Kaplan-Meier model and significant differences between treatments were assessed with the log-rank test, utilizing the survival and survminer packages. The *V. ceranae* spore load and gene transcript levels among different treatment groups were analyzed with the Kruskal-Wallis test, followed by the post-hoc Dunn's test with Bonferroni correction. Results were plotted using GraphPad Prism software v8.0.2. In all statistical analysis, p-values below 0.05 were considered significant.

## Results

### Chemical and physical properties of GanoBee

Using an ethanol-water extraction, 20–30% ß-glucans (chitin and chitosan) and 50–60% ganoderic acid were obtained. Additionally, chemical analysis indicated the presence of the following compounds: total proteins 22.40%, available carbohydrates 49.30%, total fat 1.40%, crude fiber 3.40%, minerals such as phosphorus (P 1.60%), calcium (Ca 0.03%), magnesium (Mg 0.07%), sodium (Na 0.12%), potassium (K 0.90%), manganese (Mn 0.80%), and zinc (Zn 4.30%).

We observed that bees that fell into the feeding reservoirs containing GanoBee had difficulty escaping, as the liquid adhered to their bodies, particularly the dorsal side. This coating with the product, followed by its possible absorption, often led to their death within a few hours or days. These findings indicate that the physical properties of the GanoBee solution—particularly its slight viscosity—may have contributed to the observed increase in mortality, independent of any biological effect of the extract. The density of sugar syrup and HiveAlive™ was nearly the same, but the density of GanoBee at 0.5% and 1.0% concentration, as well as GanoBee at 1.5% concentration was reduced by 0.1 g/ml and 0.2 g/ml, respectively, compared to sugar syrup and HiveAlive™. The viscosity of sugar syrup was only 0.4 mPa s higher than HiveAlive™. All three concentrations of GanoBee had relatively low viscosity compared to sugar syrup and HiveAlive™ (S2 Table).

### Effects of different diets and *Vairimorpha ceranae* infection on the survival rate and spore load

In non-infected honey bees, the group fed with 1.5% GanoBee (GB-3) had the highest survival probability among the tested concentrations (S3 Fig). Honey bees fed with 0.5% (GB-1) and 1.0% (GB-2) GanoBee showed a significantly decreased survival probability compared to control group (p = 0.004 and p < 0.001, respectively). The survival probability of the GB-3 group did not differ significantly from that of the control group (p = 0.574). Consequently, honey bees from the treatment groups GB-3 and GBN-3 were selected for gene expression and microbiome analyses. The survival analysis of the HAN group showed a significantly higher survival rate of honey bees compared to the CN group (p < 0.001). In comparison, *V. ceranae*-infected honey bees treated with different concentrations of GanoBee (GBN-1, GBN-2, GBN-3) survived significantly less than honey bees of the CN group (p < 0.001). The low viscosity of GanoBee led to increased dripping from the Falcon tubes, resulting in higher bee mortality upon contact with the deposited droplets.

The presence of *V. ceranae* was molecularly confirmed in all honey bees experimentally infected with this microsporidium, while *V. ceranae* was undetectable in all non-infected honey bees. Based on microscopical analyses, the spore load of *V. ceranae* was very low in all infected groups and varied greatly. Due to the high standard deviation, there were no significant differences between the groups. No *V. ceranae* spores were determined in the CN group on days 3 and 6 pi, the GBN-1 group on day 3 pi, and the GBN-3 group on the final sampling day (S4 Fig).

### Effects of different diets and *Vairimorpha ceranae* infection on the gut bacterial microbiota

**Relative abundance of the gut bacterial microbiota.** A total of 10,199,760 raw reads were obtained, with an average of 339,992 reads per pooled sample (S3 Table). After filtering, the total number of reads ranged from 167,475–427,509, while the number of assembled reads varied from 165,954–425,779. The percentage of chimeric sequences ranged from 1.43% to 14.39% (average of 3.33%) for assembled sequences, with a minimum length of 134 bp. The taxonomic assignment of the 30 pooled honey bee samples produced 540 ASVs grouped into 86 families and 101 different genera (S4 Table). The abundance of families varied between the treatment groups and sampling dates, but the most abundant families included Neisseriaceae (24.42%; *Snodgrassella*), Orbaceae (23.22%; *Gilliamella* and *Frischella*), Acetobacteraceae (14.67%; *Bombella* and *Commensalibacter*), Lactobacillaceae (14.50%; *Lactobacillus*) and Rhizobiaceae (12.46%; *Bartonella apis*). The family Morganellaceae (*Providencia*) was highly abundant in the GBN-3 group on day 6 pi (50.17%) and in the CN group on all sampling days except on day 6 pi (S5 and S6 Figs).

The effects of *V. ceranae* infection and the application of different diets on the relative abundance of gut bacteria at the genus level are shown in Figs 1 and 2. The most abundant genera were *Snodgrassella* (24.4%), *Gilliamella* (17.5%), *Lactobacillus* (13.3%), *Commensalibacter* (13.1%) and Rhizobiaceae (12.5%). The genus *Snodgrassella* was highly abundant across all treatment groups, with the highest relative proportion observed in the GB-3 group on day 6 pi (50.9%) and the lowest abundance in the CN group on day 3 pi (6.3%). The GB-3 group had the highest *Gilliamella* proportion on day 3 pi (55.4%), while the CN group showed the lowest (1.8%) on the same sampling day. The GBN-3 group showed the highest relative abundance of *Lactobacillus* on day 12 pi (23.8%). The relative abundance of *Commensalibacter* was higher in the HA group, attaining its maximum abundance on day 3 pi (50,4%) compared to other groups, while the GBN-3 group had the lowest relative abundance on day 6 pi (1.3%). The relative proportion of Rhizobiaceae (*Bartonella apis*) remained very

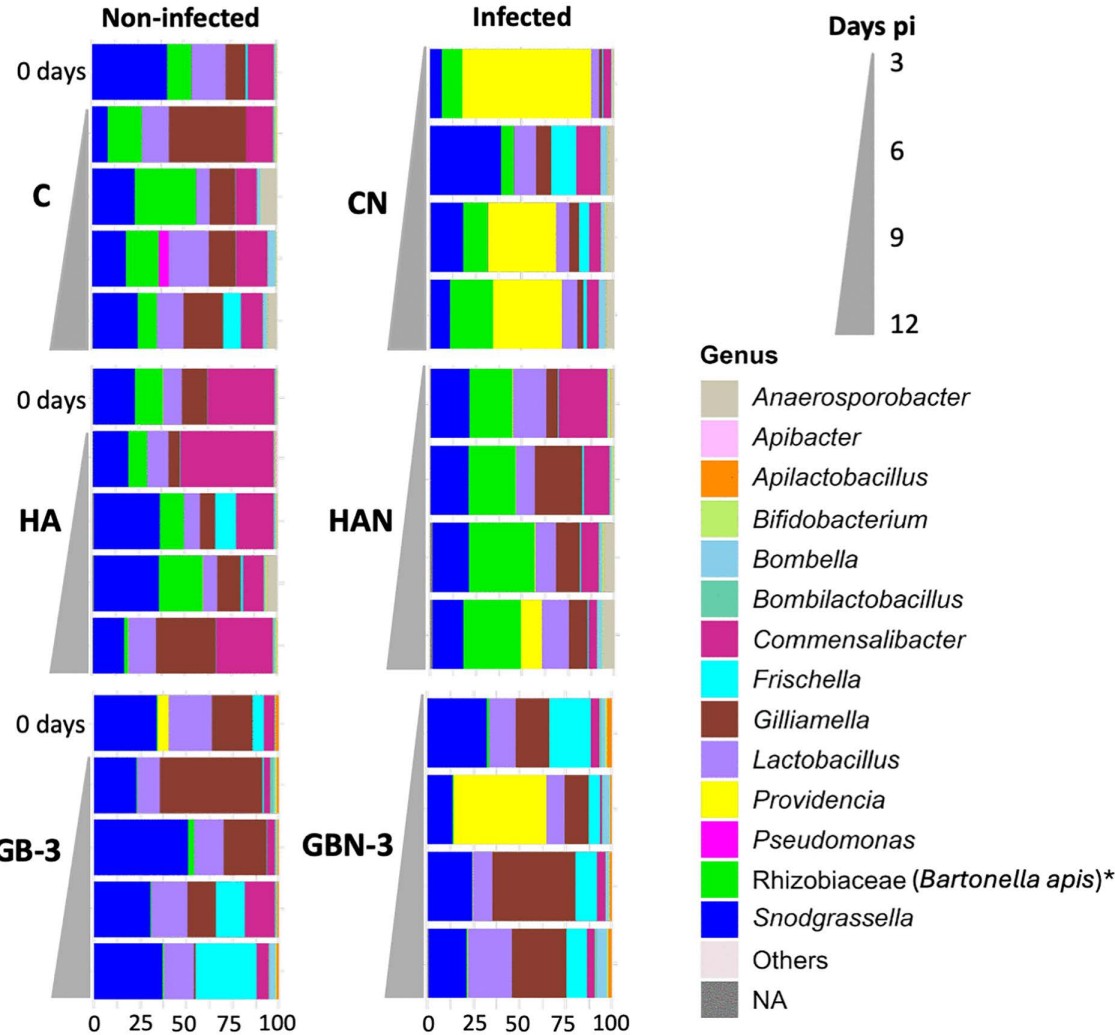

**Fig 1. Relative abundance of the gut bacterial microbiota at genus level in non-infected and *Vairimorpha ceranae*-infected honey bees under different diet conditions.** The following treatments were applied: sugar syrup (C and CN), HiveAlive™ (HA and HAN), and 1.5% GanoBee (GB-3 and GBN-3). Honey bees of the treatment groups CN, HAN and GBN-3 were infected with *V. ceranae*. Samples for analysis were collected on days 0, 3, 6, 9 and 12 post-infection (pi). Genera with an abundance of less than 0.5% were grouped under 'Other'. * Species identification by comparing the V3 - V4 regions of the 16S rRNA gene with reference sequences from GenBank database.

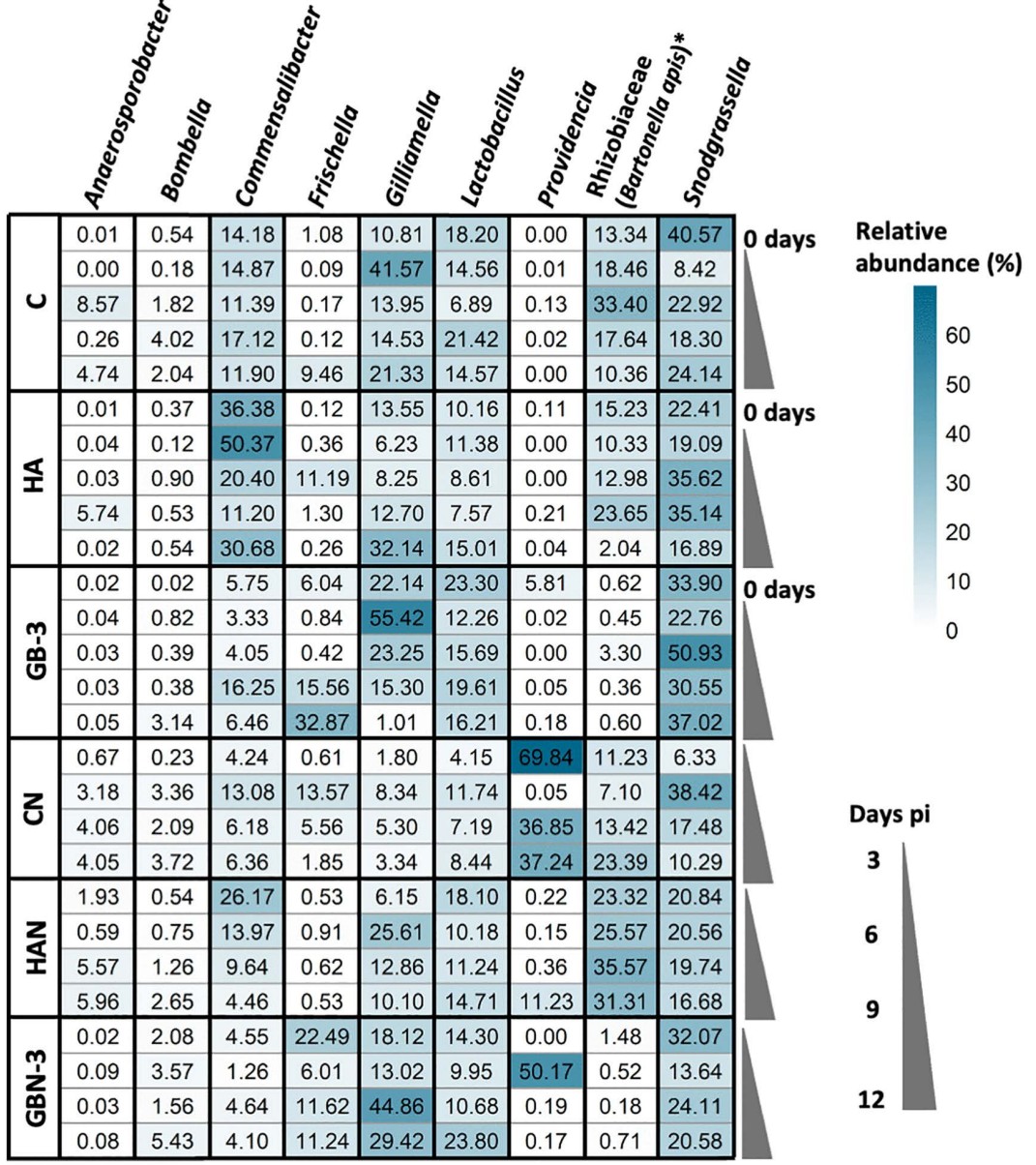

**Fig 2. Heatmap showing the relative proportions (%) of major genera identified in the gut bacterial microbiota of non-infected and *Vairimorpha ceranae*-infected honey bees under different diet conditions.** Honey bees were treated with sugar syrup (C and CN), HiveAlive™ (HA and HAN), and 1.5% GanoBee (GB-3 and GBN-3). Honey bees of the treatment groups CN, HAN, and GBN-3 were infected with *V. ceranae* on the 7th day post-emergence. Samples for analysis were taken on days 0, 3, 6, 9 and 12 post-infection (pi). * Species identification by comparing the V3 - V4 regions of the 16S rRNA gene with reference sequences from GenBank database.

low in GB-3 and GBN-3 groups across all sampling days compared to the other treatment groups. The HAN group consistently showed a high abundance of Rhizobiaceae (*Bartonella apis*) over time, reaching its maximum on day 9 pi (35%). The relative abundance of *Frischella* was low in all treatment groups except for honey bees treated with GanoBee. The highest abundance of *Frischella* was observed in the GB-3 group on day 12 pi (32.9%), followed by the GBN-3 group on day 3 pi (22.5%) and the GB-3 group again on day 9 pi (15.6%). *Providencia* was the most abundant genus in the GBN-3

group on day 6 pi (50.17%) and in the CN group across all sampling days except day 6 pi. In contrast, all non-infected treatment groups showed a very low proportion of *Providencia* at all time points.

**Alpha diversity.** All non-infected honey bees showed reduced alpha diversity on day 3 pi, but by day 9 pi it had risen sharply again (Fig 3a). The alpha diversity of *V. ceranae*-infected honey bees treated with GanoBee increased over time (Fig 3b). Differences in the mean diversity index among the different treatment groups were analyzed using the Kruskal-Wallis test (Fig 3c) but no significant differences were observed (p = 0.156 and p = 0.188 for the Shannon and Simpson diversity indices, respectively).

**Beta diversity.** PERMANOVA analysis revealed that beta diversity of GB-3 treatment group differed significantly from all other treatment groups, except for GBN-3 group (Bray-Curtis p = 0.173, Jaccard p = 0.069; Fig 4a-b). Additionally, the bacterial community composition of honey bees of the GBN-3 group was significantly different compared to all other treatment groups. However, for the Bray-Curtis dissimilarity matrix, beta diversity between the GBN-3 and CN groups was not significantly different (Bray-Curtis p = 0.065; Fig 4a). Bacterial community composition varied significantly between the CN treatment group and both the C and the HA groups (Bray-Curtis p = 0.014, Binary Jaccard p = 0.025 and Bray-Curtis p = 0.016, Binary Jaccard p = 0.019, respectively). In terms of the Binary Jaccard index (presence/absence), beta diversity differed significantly between the control and the HAN groups (Binary Jaccard p = 0.019; Fig 4b). However, beta dispersion based on Bray-Curtis dissimilarity and Binary Jaccard index was not significantly different between treatment groups (Bray-Curtis F = 0.735 p = 0.605, Binary Jaccard F = 0.806 p = 0.558) (see S5 Table).

**Effects of different diets and *Vairimorpha ceranae* infection on the gene expression of AMPs and vitellogenin**

**Comparison of gene expression among treatment groups.** On the first sampling day, the abaecin gene expression was significantly upregulated in honey bees of the HA (p = 0.040) and GB-3 (p = 0.003) groups compared to those of the control group (Fig 5a and S7 Fig). Honey bees of the GB-3 group showed a higher hymenoptaecin gene expression than the control group on the first sampling day, nearing the threshold for statistical significance (p = 0.052, Fig 5c). On the remaining sampling days, no significant differences in hymenoptaecin gene expression between the groups were found. The immune gene encoding vitellogenin was expressed at low levels across all treatment groups, with gene expression decreasing over time and no statistically significant changes observed (Fig 5e). Abaecin gene expression was increased in honey bees of the GBN-3 group compared to those of the CN and HAN groups on day 6 pi, with differences approaching statistical significance (p = 0.052 and p = 0.087, respectively) (Fig 5b). Honey bees of the GBN-3 group showed significantly higher abaecin gene expression compared to those of the CN and HAN groups on day 9 pi (p = 0.030 and p = 0.042, respectively). The highest abaecin gene expression across all treatment groups was observed in the HAN treatment group on the final sampling day. For hymenoptaecin gene expression, honey bees of the GBN-3 group showed a significant upregulation compared to the HAN group on day 6 pi (p = 0.004; Fig 5d). Gene expression of vitellogenin was significantly reduced in the CN group in comparison to the GBN-3 group on day 3 pi (p = 0.016; Fig 5f). Compared to the other two immune genes tested in this study, vitellogenin gene expression was very low across all treatment groups (see S6 Table).

**Gene expression within the same treatment group across sampling days.** When comparing abaecin gene expression within the same treatment group across the sampling days, significant differences were found in the control group between day 0 pi and day 12 pi (p = 0.012; Fig 6a). Honey bees in the GB-3 group showed a consistent gene expression of abaecin across all sampling days, whereas abaecin levels in the control and the HiveAlive™ group varied depending on the sampling day. Regarding hymenoptaecin gene expression, a significant increase was observed in the HA group from the first to the last sampling day (p = 0.012; Fig 6c). The vitellogenin gene expression steadily decreased in all non-infected honey bees from the first to the last sampling day, but no significant differences within the same treatment group were observed (Fig 6e).

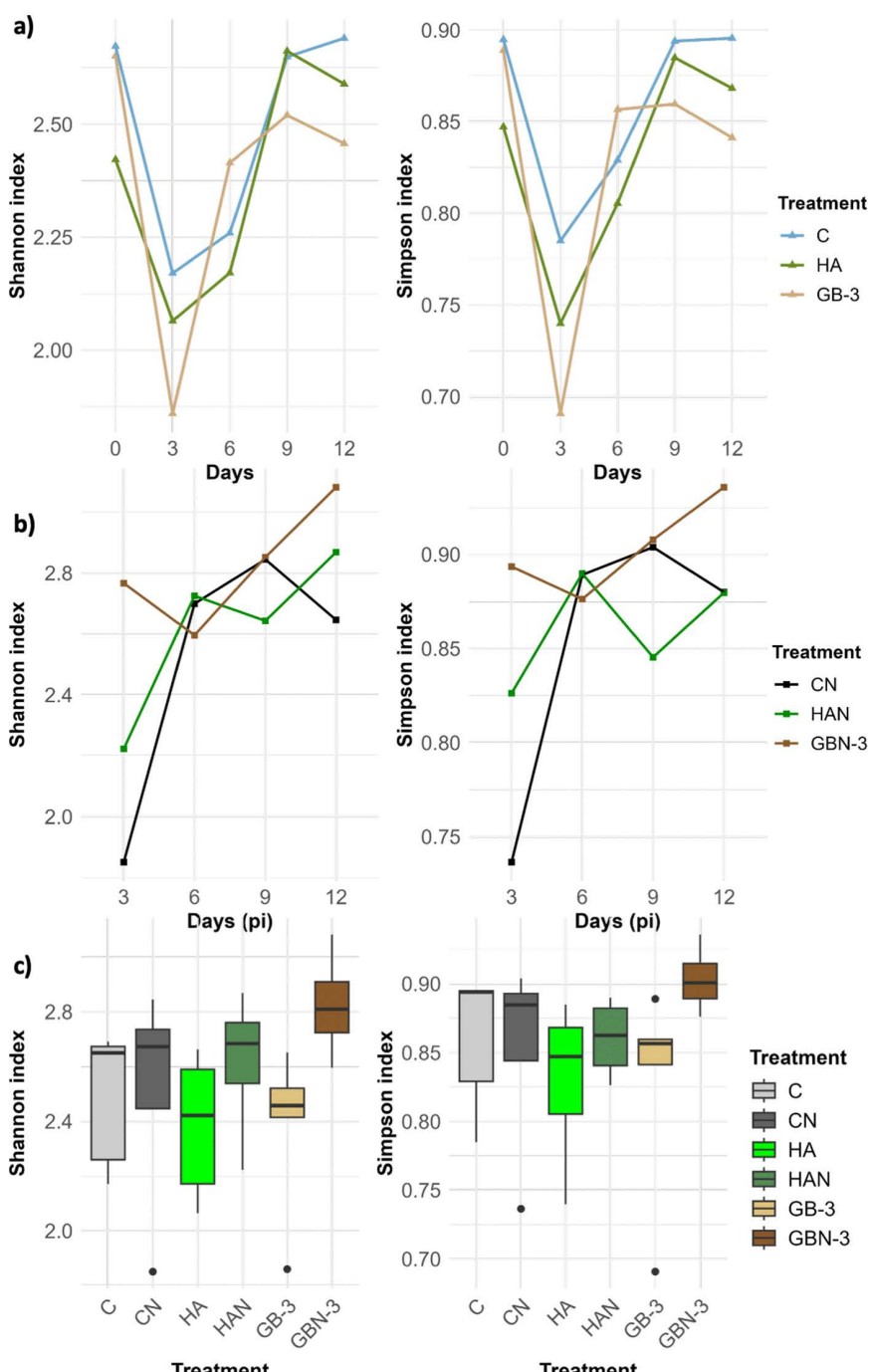

**Fig 3. Alpha diversity (Shannon and Simpson indices) of the gut bacterial microbiota in non-infected and *Vairimorpha ceranae*-infected honey bees under different diet conditions.** Three diet conditions were applied starting from the emergence of honey bees: sugar syrup (C and CN), HiveAlive™ (HA and HAN), and 1.5% GanoBee (GB-3 and GBN-3). The treatment groups CN, HAN, and GBN-3 were infected with *V. ceranae* on the 7th day post-emergence. Shannon and Simpson diversity indices in **a)** non-infected and **b)** *V. ceranae*-infected honey bees on each sampling day. **c)** Box plots represent the mean diversity index value for each treatment group (n = 4-5).

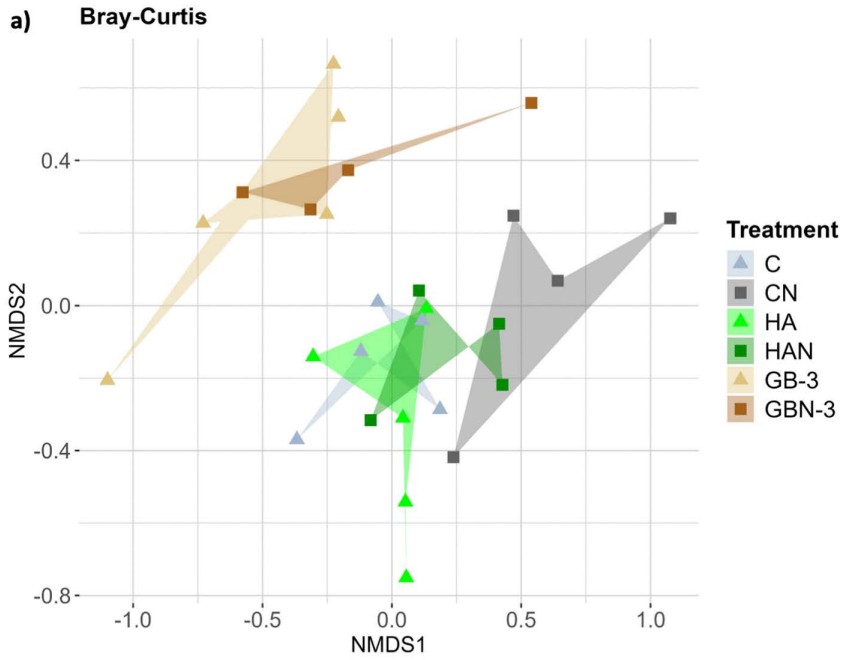

**a)** **Bray-Curtis**

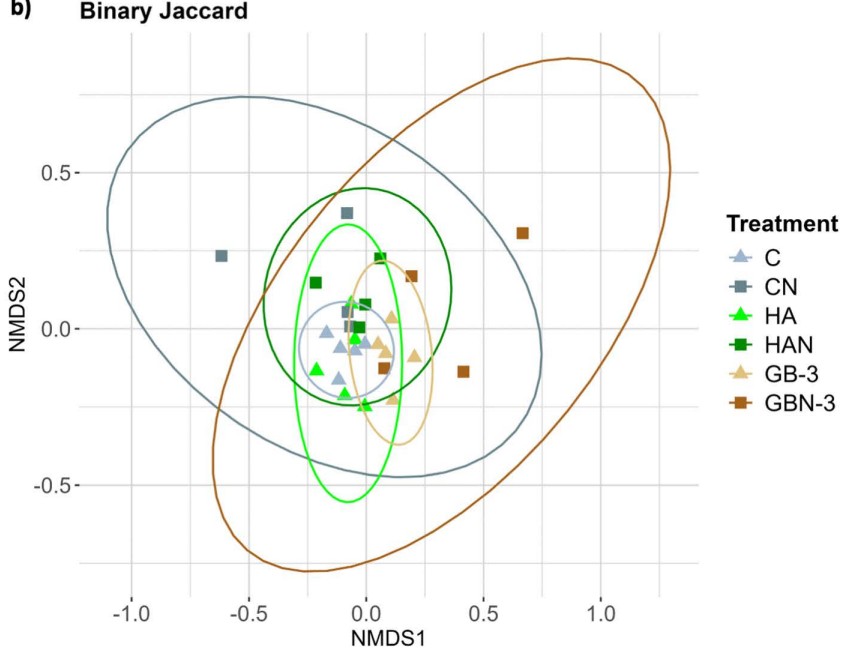

**b)** **Binary Jaccard**

**Fig 4. Non-metric Multidimensional Scaling (NMDS) plots illustrate beta diversity of the gut bacterial composition among honey bee treatment groups.** Honey bees were fed with sugar syrup (C and CN), HiveAlive™ (HA and HAN), and 1.5% GanoBee (GB-3 and GBN-3). The treatment groups CN, HAN, and GBN-3 were infected with *Vairimorpha ceranae* seven days after emergence. NMDS analysis was based on **a)** Bray-Curtis dissimilarity and **b)** Jaccard's coefficient for binary data. Statistical differences were assessed with Permutational Multivariate Analysis of Variance (PERMANOVA). The ellipses show 95% confidence intervals per treatment group.

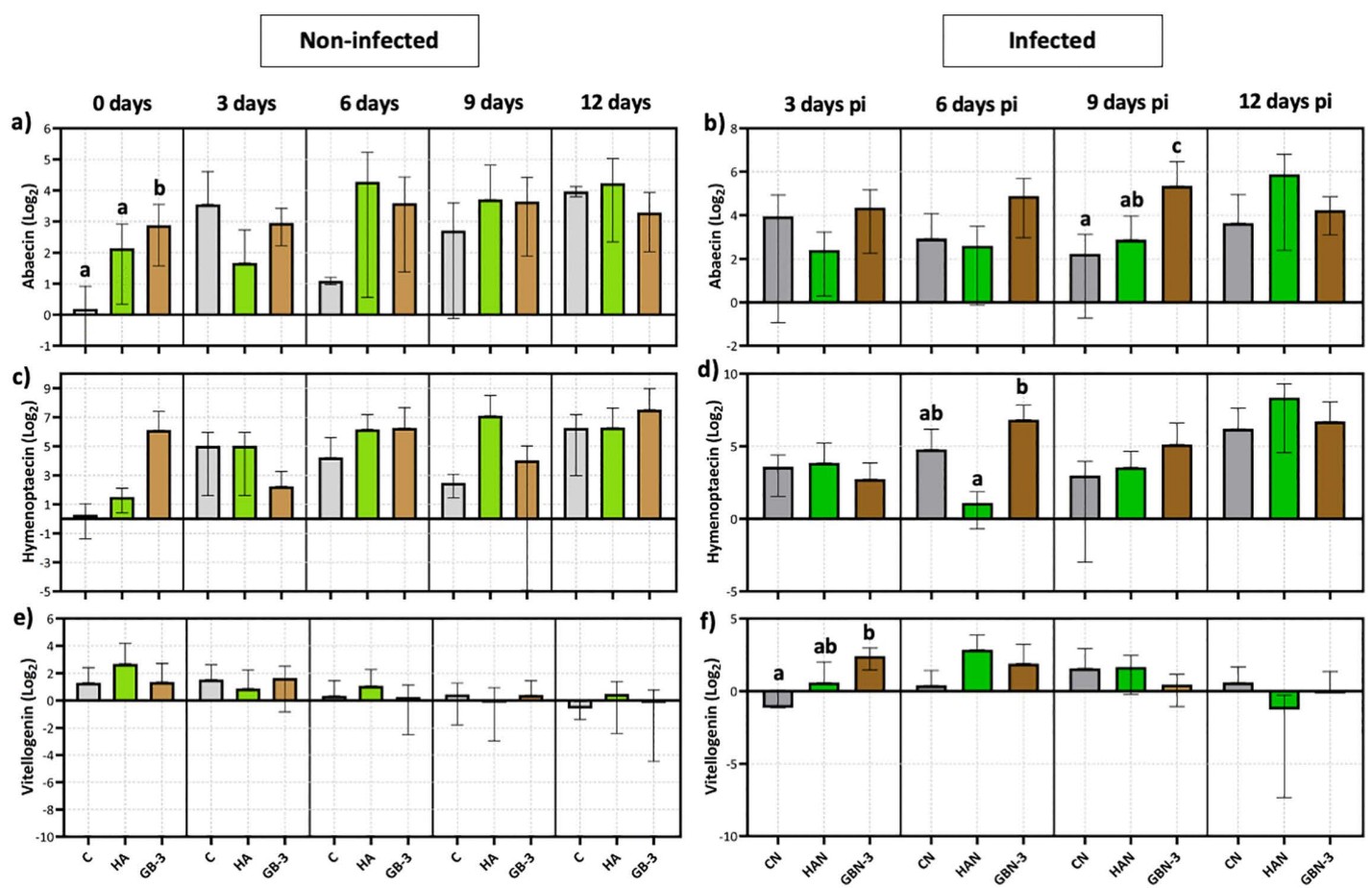

**Fig 5. Effects of applied diets and *Vairimorpha ceranae* infection on the relative immune gene expression in honey bees.** Graphs show relative gene expression ($2^{-\Delta\Delta Ct}$) of abaecin **(a-b)**, hymenoptaecin **(c-d)** and vitellogenin **(e-f)** (non-infected n = 3 and infected n = 5−7). Honey bees were sampled for analysis on days 0, 3, 6, 9 and 12 post-infection (pi). Three different diets were applied from the beginning of emergence: sugar syrup (C and CN), HiveAlive™ (HA and HAN) and 1.5% GanoBee (GB-3 and GBN-3). The treatment groups CN, HAN, and GBN-3 were infected with *V. ceranae* on the 7th day post-emergence. Different letters indicate statistically significant differences between treatment groups.

The abaecin gene expression in *V. ceranae*-infected honey bees did not vary significantly within the same treatment group across the sampling days (Fig 6b). Honey bees in the HAN group showed a constant abaecin gene expression from day 3 pi until day 9 pi but abaecin levels increased on the last sampling day. The GBN-3 group exhibited a constant abaecin gene expression across all sampling days. Honey bees in the HAN group showed a significant increase of hymenoptaecin gene expression from day 6 pi to the last sampling day (p = 0.004; Fig 6d). In the CN group, only a slight variation in the hymenoptaecin gene expression was determined among the sampling days. A significant reduction of the vitellogenin gene expression was noted in the HAN group from day 6 pi to day 12 pi (p = 0.009) and in the GBN-3 group from day 3 pi to day 12 pi (p = 0.037; Fig 6f) (see S6 Table).

**Correlation analyses between the relative abundance of gut bacteria and immune-related gene expression**

Correlation analyses were conducted between the relative abundance of gut bacteria at the genus level and gene expression of abaecin, hymenoptaecin, and vitellogenin. However, no pairwise correlations were found to be statistically significant (S7 Table).

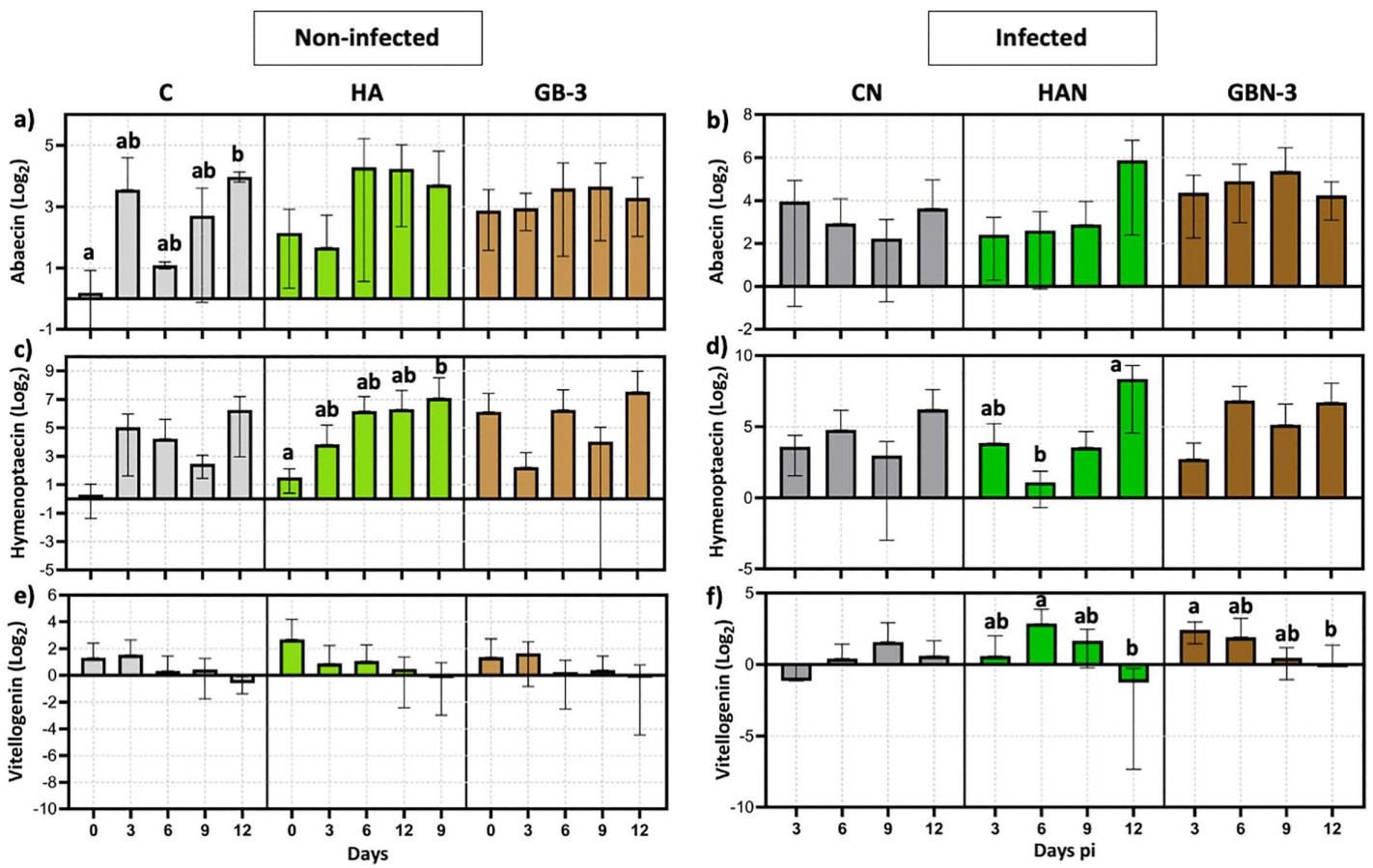

**Fig 6. Effects of different diets and *Vairimorpha ceranae* infection on the relative immune gene expression in honey bees within the same treatment group.** Graphs show relative gene expression ($2^{-\Delta\Delta Ct}$) of abaecin **(a-b)** hymenoptaecin **(c-d)** vitellogenin **(e-f)** in honey bees (non-infected: n = 3 and infected: n = 5−7). Honey bees were sampled for analysis on days 0, 3, 6, 9 and 12 post-infection (pi). Three different diets were applied from the beginning of emergence: sugar syrup (C and CN), HiveAlive™ (HA and HAN) and 1.5% GanoBee (GB-3and GBN-3). The treatment groups CN, HAN and GBN-3 were infected with *V. ceranae* on the 7th day after emergence. Different letters indicate statistically significant differences between treatment groups.

## Discussion

Our study investigated the effects of the dietary supplement GanoBee on gut bacterial microbiota composition and immune-related gene expression in both non-infected and *Vairimorpha ceranae*-infected honey bees. The findings revealed that the gut microbiota across all treatment groups was predominantly composed of core bacterial genera, including *Snodgrassella*, *Gilliamella*, and *Lactobacillus*, aligning with observations from previous studies [61,62]. However, the relative abundance of different bacterial genera varied depending on feeding type and *V. ceranae* infection. Honey bees of the GB-3 group showed an increased abundance of *Gilliamella* and *Frischella* and a decreased abundance of *Commensalibacter* and Rhizobiaceae (*Bartonella apis*) compared to those of the C and HA groups. These results strengthen previous findings that different dietary supplements alter the bacterial microbiota of honey bees [21,22,62].

Overall, a low *Vairimorpha* infection level was observed, possibly influenced by spore storage conditions [63,64], environmental factors during cage trials [65], or the age of bees at spore inoculation [66–68]. The CN group showed increased *Providencia* and reduced *Lactobacillus* abundance compared with HAN and GBN-3. *Providencia* spp., common in various

insects including honey bees [69] and pathogenic in *Drosophila melanogaster* [70], may have increased in response to *V. ceranae* infection, as this genus is low in non-infected bees. The reduction in *Lactobacillus* is consistent with previous reports on *V. ceranae*-infected honey bees [71,72]. Moreover, GanoBee-fed bees exhibited lower *Rhizobiaceae* (*Bartonella apis*) and higher *Frischella* abundance. The species *F. perrara* is associated with the scab phenotype in the pylorus [73], but its role in honey bees immune system remains controversial: while aged pollen consumption increased *F. perrara* abundance and mortality [74], experimental colonization has been shown to stimulate immune activation [75].

Previous studies [26,61] confirmed the observation that *V. ceranae* infection did not affect the alpha diversity of the honey bee gut bacteriome. In contrast, Lau et al. [27] showed that alpha diversity significantly differed between *V. ceranae*-infected and control bees, although their study involved bees that did not emerge in a laboratory setting, unlike the present study. In addition, previous investigations have shown that different dietary supplements affected alpha diversity in honey bees [23,24].

Significant differences in beta diversity were observed between the GB-3 and GBN-3 groups compared with all other treatments, except between CN and GBN-3, based on Bray–Curtis dissimilarity. These differences were not associated with changes in beta dispersion, indicating consistent within-group variability. No significant difference was found between GB-3 and GBN-3, suggesting that the observed shifts were mainly driven by the GanoBee diet rather than *V. ceranae* infection. The bacterial community composition differed significantly between non-infected, and *V. ceranae*-infected bees fed with sugar syrup, consistent with previous studies showing that *V. ceranae* alters gut microbiota composition [25,26]. We propose that GanoBee provided a stabilizing effect on the gut microbiota of infected bees, potentially preventing dysbiosis and maintaining microbial balance. Similarly, Huang et al. [24] demonstrated that diet influences gut microbiota composition in *V. ceranae*-infected bees. GanoBee is rich in polysaccharides, particularly β-glucans such as chitin and chitosan, which have been shown to modulate gut microbiota and reduce inflammation [39]. Moreover, *Ganoderma* polysaccharides possess antimicrobial properties [38,76], which may explain the distinct bacterial composition in GanoBee-fed bees and the absence of infection-related shifts in their beta diversity.

The upregulation of AMP gene expression in honey bees in response to different diets [20,71] can be confirmed in this study. Honey bees of the GB-3 group showed consistently high abaecin gene expression over time with a significant increase on the first sampling day. The hymenoptaecin gene expression significantly increased within the HA group from the first to the last sampling day. However, the vitellogenin gene expression decreased in all treatment groups over time, with no significant differences between them. Similarly, Garrido et al. [32] found no significant changes in midgut vitellogenin gene expression in honey bees fed with different bacterial strains. Honey bees treated with essential oils and propolis extracts showed consistent vitellogenin gene expression among all treatment groups [21]. However, diets containing pollen substitutes led to an increase in vitellogenin gene expression in honey bees compared to a pollen-free diet [22,23].

In this study, the gene expression of AMPs and vitellogenin in *V. ceranae*-infected honey bees varied depending on the applied diet. The effects of *V. ceranae* infection on the expression of immune-related genes are not completely understood. However, various studies have reported that infection with *V. ceranae* suppresses the gene expression of AMPs and vitellogenin in honey bees, although the experimental designs and timing of *V. ceranae* inoculation varied across studies [12–15]. The upregulation of abaecin and hymenoptaecin gene expression in honey bees fed with GanoBee indicates that this dietary supplement stimulates the immune system, thereby mitigating the immunosuppressive effects of *V. ceranae* infection.

The hypothesis that fungal extracts enhance the immune system of honey bees and provide protection against parasites is supported by previous studies [34–36,77]. However, in contrast to Glavinic et al. [35,36] the effects of the GanoBee diet on the immune-related gene expression were analyzed over a longer period, which may explain the observed decrease in vitellogenin gene expression over time in this study. The upregulation of immune-related gene expression in honey bees fed with GanoBee may be explained by the high content of polysaccharides and ganoderic

acids contained in the *G. australe* extract. Several studies have shown the anti-inflammatory, antioxidant, and immuno-modulatory activities of the bioactive components of *Ganoderma* [37,40,41,78].

An important limitation of this study was the unexpectedly high mortality observed in certain treatment groups, including uninfected controls, and particularly those supplemented with GanoBee. Post hoc analyses indicated that the GanoBee-supplemented diets exhibited greater viscosity than the other feeding solutions. Consequently, bees that accidentally fell into the feeders were often unable to escape, leading to rapid mortality. These findings suggest that the physical properties of the GanoBee formulation, rather than its biological activity alone, may have contributed to the elevated mortality rates. Previous studies have similarly reported that feeder design and the physical characteristics of feeding solutions can significantly influence honey bee survival under laboratory conditions [79]. Therefore, the feeding system itself should be recognized as a critical experimental factor. Future investigations employing similar conditions should consider optimizing the GanoBee formulation—by reducing viscosity or adapting the feeder design—to minimize unintended mortality and allow for a more accurate assessment of its biological effects.

A significant correlation between gut bacterial genera and immune-related gene expression was not detected in this study. However, Ewert et al. [21] showed a positive correlation between the vitellogenin gene expression and the abundance of *Lactobacillus* Firm-4 in honey bees, but no correlation was observed between the abaecin gene expression and any of the tested gut bacteria. A study by Kim et al. [23] demonstrated a negative correlation between vitellogenin gene expression levels and the abundance of *Gilliamella*.

This is the first study that demonstrates the effects of a *Ganoderma australe* extract on gut bacterial microbiota and immune-related gene expression in honey bees under controlled laboratory conditions. GanoBee supplementation appeared to prevent *Vairimorpha*-induced dysbiosis and enhanced antimicrobial peptide gene expression, indicating its potential as a dietary supplement to support bee health. However, *Vairimorpha ceranae* infection levels were unexpectedly low, likely due to spore storage or environmental factors in cage-based assays, which may have limited the assessment of GanoBee's protective effects. Elevated mortality in some groups was attributed to the high viscosity of the GanoBee formulation rather than its biological activity. These findings underscore the need for optimized infection protocols and feeding systems. With further optimization, GanoBee could be safely and effectively applied in future research and apicultural practice.

## Supporting information

**S1 Fig. Collection site for fruiting body of *Ganoderma australe* used in obtaining pure cultures. a)** Location in Chile of the collection site marked with a red square. **b)** Collection site in Fundo Teja Norte. OpenStreetMap contributors (2015); map retrieved in December 2024 from https://planet.openstreetmap.org. **c)** Fruiting body of *Ganoderma australe* growing on a living *Nothofagus obliqua* tree.
(TIF)

**S2 Fig. GanoBee product and feeding of honey bees under experimental conditions. a)** GanoBee product containing 1.5% *Ganoderma australe* mycelial extract in a 50% sugar syrup solution with co-formulate. **b)** Experimental cage for honey bees with a water feeder (Eppendorf tube) and a feeding device containing GanoBee (Falcon tube).
(TIF)

**S3 Fig. Effects of different diets and *Vairimorpha ceranae* infection on the survival rate of honey bees. a)** Survival rates of non-infected honey bees exposed to different diets — sugar syrup (C), HiveAlive™ (HA) and GanoBee at three different concentrations (GB-1, GB-2 and GB-3). **b)** Survival rates of honey bees infected with *V. ceranae* seven days post- emergence and exposed to different diets—sugar syrup (CN), HiveAlive™ (HAN) and GanoBee at three different concentrations (GBN-1, GBN-2 and GBN-3). Survival rates were analyzed using the Kaplan-Meier model.
(TIF)

**S4 Fig. *Vairimorpha ceranae* spore loads in infected treatment groups.** Honey bees were fed either sugar syrup (CN), HiveAlive™ (HAN), or GanoBee at three different concentrations (GBN-1, GBN-2 and GBN-3) on the first day after emergence until the end of the experiment. Honey bees were infected with *V. ceranae* on the 7th day post-emergence. The spore loads of the honey bees sampled at 3-, 6-, 9- and 12-days post-infection (pi) were expressed as means ± S.D. (n = 4–5). N.O. means no spore load was observed.
(TIF)

**S5 Fig. Relative abundance of the gut bacterial microbiota at family level in non-infected and *Vairimorpha ceranae*-infected honey bees under different diet conditions.** The following diets were applied: sugar syrup (C and CN), HiveAlive™ (HA and HAN), and 1.5% GanoBee (GB-3 and GBN-3). Honey bees of the treatment groups CN, HAN and GBN-3 were infected with *V. ceranae* on the 7th day post-emergence. Samples for analysis were taken on days 0, 3, 6, 9 and 12 post-infection (pi). Families with an abundance of less than 0.5% were grouped under 'Others'.
(TIF)

**S6 Fig. Heatmap showing the relative proportions (%) of major families identified in the gut bacterial microbiota of non-infected and *Vairimorpha ceranae*-infected honey bees under different diet conditions.** Honey bees were treated with sugar syrup (C and CN), HiveAlive™ (HA and HAN) and 1.5% GanoBee (GB-3 and GBN-3). Honey bees of the treatment groups CN, HAN and GBN-3 were infected with *V. ceranae*. Samples for analysis were taken on days 0, 3, 6, 9 and 12 post-infection (pi).
(TIF)

**S7 Fig. Heatmap of relative immune gene expression across treatment groups on different sampling days.** The $Log_2$ values of relative expression ratios of abaecin, hymenoptaecin and vitellogenin between non-infected and *Vairimorpha ceranae*-infected treatment groups are demonstrated. Three diets were applied from the beginning of emergence: sugar syrup (C and CN), HiveAlive™ (HA and HAN) and 1.5% GanoBee (GB-3 and GBN-3). The treatment groups CN, HAN and GBN-3 were inoculated with *V. ceranae* on the 7th day post-emergence. Honey bees were sampled for analysis on days 0, 3, 6, 9 and 12 post-infection (pi). NA means no sampled honey bees. Asterisks show statistically significant differences between treatment groups: * $p < 0.05$; ** $p < 0.01$.
(TIF)

**S1 Table. Primers of target genes related to the immune response of *Apis mellifera* used for RT-qPCR.**
(DOCX)

**S2 Table. Density and viscosity of dietary applications used in honey bee feeding.** The density and viscosity of sugar syrup, HiveAlive™ and three different concentrations of GanoBee (0.5%, 1.0% and 1.5%) were determined in triplicate at 30°C. The water density of 0.9956 $g/cm^3$ and the viscosity of 0.7972 mPa s were used for the analyses.
(XLSX)

**S3 Table. Summary of the Illumina sequencing analysis with a total of 30 pooled honey bee samples.**
(XLSX)

**S4 Table. Taxonomic assignment of gut bacterial OTUs from honey bees subjected to different treatments, as detailed in the methodology section.** The dataset includes OTU sequences, sample information, relative abundance, treatment groups, evaluation days post-treatment, and taxonomic classification at the phylum, class, order, family, and genus levels.
(XLSX)

**S5 Table. Permutational Multivariate Analysis of Variance (PERMANOVA) assessing statistical differences in the gut bacterial composition among treatment groups, based on Bray-Curtis dissimilarity and the Binary Jaccard index.** The treatment comparisons are listed in the left-hand column. Asterisks indicate statistically significant differences: * p<0.05; ** p<0.01.
(XLSX)

**S6 Table. Raw gene expression data for abaecin, hymenoptaecin, and vitellogenin in non-infected and *Vairimorpha ceranae*-infected honey bees.** Data are distributed across three sheets corresponding to each gene in the Excel file.
(XLSX)

**S7 Table. Correlation analyses between the relative abundance of gut bacterial genera and immune-related gene expression in both non-infected and Vairimorpha ceranae-infected honey bees.** The correlation between the relative abundance of gut bacteria at the genus level and the gene expression of abaecin, hymenoptaecin, and vitellogenin in both non-infected and V. ceranae-infected honey bees were analyzed, using the non-parametric Spearman rank correlation test. The p-value was adjusted with a Bonferroni correction for multiple comparisons. The strength and direction of the relationship (negative and positive) were evaluated based on Spearman's rho ($\rho$) values (scale from −1–1).
(XLSX)

## Acknowledgments

We would like to thank L. Silvestre for its assistance with honey bee fieldwork and C. Oyarzún for HPLC analyses. We also thank Dr. Suwannapong and an anonymous reviewer for the useful comments, which have helped to improve our manuscript.

## Author contributions

**Conceptualization:** Sigisfredo Garnica.

**Formal analysis:** Sarah Zuern, Bella Romero, Carlos Spichiger, Max Emil Schön.

**Funding acquisition:** Esteban Basoalto, Max Emil Schön, Sigisfredo Garnica.

**Investigation:** Sarah Zuern, Sigisfredo Garnica.

**Methodology:** Sarah Zuern, Leandro Ortiz, Alejandro Jerez, Sigisfredo Garnica.

**Project administration:** Sigisfredo Garnica.

**Resources:** Esteban Basoalto, Max Emil Schön, Sigisfredo Garnica.

**Writing – original draft:** Sarah Zuern, Sigisfredo Garnica.

**Writing – review & editing:** Bella Romero, Alejandro Jerez, Esteban Basoalto, Max Emil Schön.

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
