## [Decision Letter · Decision Letter 0]

13 Oct 2025

Dear Dr. Garnica,

Thank you for submitting your manuscript to PLOS ONE. After careful consideration, we feel that it has merit but does not fully meet PLOS ONE’s publication criteria as it currently stands. Therefore, we invite you to submit a revised version of the manuscript that addresses the points raised during the review process.

We look forward to receiving your revised manuscript.

Kind regards,

Kai Wang

Academic Editor

PLOS ONE

Journal Requirements:

“Funding for this project was made possible by project the Applied Research and Innovation 2023 grant INID210009. ME. Schön would like to thank I. Schlichting and the Max Planck Society for their support.”

Reviewer's Responses to Questions

**Comments to the Author**

1. Is the manuscript technically sound, and do the data support the conclusions?

Reviewer #1: Yes

Reviewer #2: Partly

2. Has the statistical analysis been performed appropriately and rigorously?

Reviewer #1: Yes

Reviewer #2: Yes

3. Have the authors made all data underlying the findings in their manuscript fully available?

Reviewer #1: Yes

Reviewer #2: Yes

4. Is the manuscript presented in an intelligible fashion and written in standard English?

Reviewer #1: Yes

Reviewer #2: Yes

Reviewer #1: After I looked through the manuscript titled “Exploratory study on the impact of Ganoderma australe extract on gut microbiota and immune gene expression in honey bees exposed to Nosema ceranae”; Number PONE-D-25-39736. The manuscript looks fair to me; however, the manuscript needs to be revised as a major revision. Please see the comments below.

Abstract:

Looks good to me; however, I suggest the authors add the spore load (number of spores per bee).

Keywords: looks good to me.

Introduction:

Page 11: lines 35: “xxxx first described in the Asian honey bee (Apis ceranae)”

Please replace Apis ceranae with Apis cerana

Materials and methods:

Page 14, Lines 119-120: “The fungal material was macerated using a sterile pistil and 300 μl buffer was added.”

Please provide what buffer the authors used for this one.

Line 122: “xxx of 5 min each in a Mini-BeadBeater-16 at 3450 rpm.”

Please convert 3450 rpm to xxx g

xg (RCF) = (RPM)2 x radius* x 1.118 x 10-5

*The distance of particles from the center of rotation (centrifuge radius), unit in centrimeter (cm)

Please also provide the model and brand of the centrifuge the authors used.

Page 15, Lines 139-140: “Fungal culture was grown for three weeks in laboratory conditions.”

Please provide the actual laboratory condition (e.g., temperature, nutrients, humidity, sterile environment.

Line 147: “using high-performance liquid chromatography (HPLC)”

Please provide the model, brand, and the condition, e.g., Detector type, xxx nm; the Column? (e.g., Cxx, xx mm × xx mm, x µm column), temperature xx °C; the mobile phase consisted of Solvent x: Solvent y

The flow rate of xx mL/min in xx mode; The injection volume was xx µL.

The software name used for data acquisition and analysis.

Line 151-153: “ Only hives that were vigorous and showed no symptoms of any disease were used in the experiments”

I am wondering how the authors ensure that those five colonies are free of pathogens. Please provide the methods the authors used to confirm that they are disease-free colonies.

Page 16, Line 170-171: “On the 7th day post-emergence five treatment groups were infected with Nosema ceranae.”

Please provide the dose of Nosema ceranae, how many spores per bee the authors used for this experiment.

Page 17, Lines 198-200: “Seven days after emergence, honey bees from the treatment groups CN, HAN, GBN-1, GBN-2, and GBN-3 were infected with N. ceranae spores. Honey bees were fed two times a 2 ml spore solution. Once the total infection dose was consumed,”

What exactly number of N. ceranae spores used for each group? 2x 106 spores per group? By group feeding? If so, please add more detail.

Lines 201-202: “xxx five honey bees of each cage were fed individually 10 µl of N. ceranae spore solution using a micropipette according to the protocol described by Williams et al. [54].”

How many spores were in 10 µl that the authors fed each bee?

Results:

Page 21, line 317: “xxx GanoBee at 1.5% concentration was reduced by 0.1 g/cm³ and 0.2 g/cm³, “

I suggest the authors provide the unit of concentration as g/ml instead of g/cm3

Line 321: “Effects of different diets and N. ceranae infection on thexxx “

I suggest the authors replace N. ceranae with Nosema ceranae since it is a subtopic

Page 22, Line 343-344: “Effects of different diets and N. ceranae infection on the gut bacterial

microbiota ”

Please replace N. ceranae with Nosema ceranae

Figure 3: Please provide a better solution for the graphs and figure legend of all statistics

Figures 5 and 6: Please provide the different letters showing the statistical difference in the data in Figure 5 and 6 instead of * or **

Discussion: I suggest that the authors improve the discussion part by making it shorter, emphasizing, and

clarifying the scientific contribution of this paper.

References: Please take care of scientific names, make sure they are in italic form.

Supporting information: Please take care of scientific names, make sure they are in italic form.

Reviewer #2: 1. The Nosema ceranae is now considered as Vairimorpha ceranae, therefore, the author should correct this genus everywhere in this manuscript.

2. The major concern of this manuscript is the author mentioned the establishment of N. ceranae infection appeared limited, likely due to low spore viability. In my view point, the microsporidia infection in honey bee should be the basic experiment of this study, which the author should overcome this problem before they trying to publish this paper, otherwise the V. ceranae infection test should be removed in this manuscript and focus on the Ganoderma australe extract on gut microbiota and immune gene expression in honey bees

3. Table 1: The standard V. ceranae infection process should observe more than 2-weeks (14~21days) and therefore can see the pathogenesis of V. ceranae.

4. The flowchart of bioinformatics could be submitted to GitHub system

**Do you want your identity to be public for this peer review?** For information about this choice, including consent withdrawal, please see our Privacy Policy

Reviewer #1: **Yes:** Guntima Suwannapong

Reviewer #2: No

---

## [Author Response · Author response to Decision Letter 1]

11 Dec 2025

PONE-D-25-39736

Exploratory study on the impact of Ganoderma australe extract on gut microbiota and immune gene expression in honey bees exposed to Nosema ceranae

In the following pages, we give the details of all changes/improvements to the revised version of our manuscript.

We have improved our manuscript (incl. figures) following the PLOS ONE staff’ recommendations and reviewers (Please see “Point-by-point response to reviewers”).

Sincerely,

Sigisfredo Garnica

PONE-D-25-39736

Exploratory study on the impact of Ganoderma australe extract on gut microbiota and immune gene expression in honey bees exposed to Nosema ceranae

PLOS ONE

Dear Dr. Garnica,

Thank you for submitting your manuscript to PLOS ONE. After careful consideration, we feel that it has merit but does not fully meet PLOS ONE’s publication criteria as it currently stands. Therefore, we invite you to submit a revised version of the manuscript that addresses the points raised during the review process.

We look forward to receiving your revised manuscript.

Kind regards,

Kai Wang

Academic Editor

PLOS ONE

Journal Requirements:

We have revised our manuscript to meet PLOS ONE's style requirements, including file naming.

The ORCID iD for the corresponding author was validated in Editorial Manager.

“Funding for this project was made possible by project the Applied Research and Innovation 2023 grant INID210009. ME. Schön would like to thank I. Schlichting and the Max Planck Society for their support.”

Yes, effectively the funders had no role. Please include the following sentence: The funders had no role in study design, data collection and analysis, decision to publish, or preparation of the manuscript.

Please for Data Availability change to “Raw sequencing data files are available from the Sequence Read Archive (SRA) under BioProject number PRJNA1347411 (http://www.ncbi.nlm.nih.gov/bioproject/1347411).”

There are no such comments from the reviewers.

Reviewer's Responses to Questions

Point-by-point response to reviewers.

Reviewer #1:

After I looked through the manuscript titled “Exploratory study on the impact of Ganoderma australe extract on gut microbiota and immune gene expression in honey bees exposed to Nosema ceranae”; Number PONE-D-25-39736. The manuscript looks fair to me; however, the manuscript needs to be revised as a major revision. Please see the comments below.

Abstract:

Looks good to me; however, I suggest the authors add the spore load (number of spores per bee).

We introduced the information suggested by the reviewer into the abstract.

Introduction:

Page 11: lines 35: “xxxx first described in the Asian honey bee (Apis ceranae)”

Please replace Apis ceranae with Apis cerana

We replaced Apis ceranae with Apis cerana.

Materials and methods:

Page 14, Lines 119-120: “The fungal material was macerated using a sterile pistil and 300 μl buffer was added.”

Please provide what buffer the authors used for this one.

Now we provide the Buffer used.

Line 122: “xxx of 5 min each in a Mini-BeadBeater-16 at 3450 rpm.”

Please convert 3450 rpm to xxx g

xg (RCF) = (RPM)2 x radius* x 1.118 x 10-5

*The distance of particles from the center of rotation (centrifuge radius), unit in centrimeter (cm)

The shaking motion of the Mini-BeadBeater-16 involves a "throw" or vial displacement, which is specified as 2.22 cm. This value represents the distance the vials move during agitation, not a radius of the device itself.

Please also provide the model and brand of the centrifuge the authors used.

We provide the model and brand: Mini-BeadBeater-16 Model670EUR, BioSpec Products, Inc., USA.

Page 15, Lines 139-140: “Fungal culture was grown for three weeks in laboratory conditions.”

Please provide the actual laboratory condition (e.g., temperature, nutrients, humidity, sterile environment.

We added a sentence characterizing the laboratory conditions as follows: Sterile environment, around 23°C room temperature and 49% humidity.

Line 147: “using high-performance liquid chromatography (HPLC)”

Please provide the model, brand, and the condition, e.g., Detector type, xxx nm; the Column? (e.g., Cxx, xx mm × xx mm, x µm column), temperature xx °C; the mobile phase consisted of Solvent x: Solvent y

The flow rate of xx mL/min in xx mode; The injection volume was xx µL.

The software name used for data acquisition and analysis.

The following information was introduced into the main manuscript text:

Ganoderic acids were detected with a high-performance liquid chromatography (HPLC) high-performance liquid chromatography (HPLC) analysis using an Agilent 1200 Series system equipped with a diode array detector (MWD 61365 D, Agilent). The analytical procedure was modified from Liu et al. [49]. The detection wavelength was set at 250 nm. Separation was achieved on an Agilent Zorbax 300SB column (5 µm, 4.6 mm × 250 mm), maintained at 30°C. The mobile phase consisted of a gradient elution using (A) acetonitrile and (B) 0.1% acetic acid in water (v/v). The injection volume was 20 µL, and the flow rate was set to 1.0 mL/min in continuous mode. Data acquisition and analysis were conducted using Agilent ChemStation Software.

Line 151-153: “Only hives that were vigorous and showed no symptoms of any disease were used in the experiments”

I am wondering how the authors ensure that those five colonies are free of pathogens. Please provide the methods the authors used to confirm that they are disease-free colonies.

Now we detailed the methods used to indicate that they are disease-free colonies.

Page 16, Line 170-171: “On the 7th day post-emergence five treatment groups were infected with Nosema ceranae.”

Please provide the dose of Nosema ceranae, how many spores per bee the authors used for this experiment.

The concentration solution used was 1 x 106 spores per ml. We introduced this information into the text.

Page 17, Lines 198-200: “Seven days after emergence, honey bees from the treatment groups CN, HAN, GBN-1, GBN-2, and GBN-3 were infected with N. ceranae spores. Honey bees were fed two times a 2 ml spore solution. Once the total infection dose was consumed,”

What exactly number of N. ceranae spores used for each group? 2x 106 spores per group? By group feeding? If so, please add more detail.

Similarly, we used the concentration solution of spores per group as previously described.

Lines 201-202: “xxx five honey bees of each cage were fed individually 10 µl of N. ceranae spore solution using a micropipette according to the protocol described by Williams et al. [54].”

How many spores were in 10 µl that the authors fed each bee?

We used the concentration solution of 1 x 106 spores per ml.

Results:

Page 21, line 317: “xxx GanoBee at 1.5% concentration was reduced by 0.1 g/cm³ and 0.2 g/cm³, “

I suggest the authors provide the unit of concentration as g/ml instead of g/cm3

We replaced 0.1 g/cm³ and 0.2 g/cm³ with 0.1 g/mL and 0.2 g/mL, respectively.

Line 321: “Effects of different diets and N. ceranae infection on thexxx “

I suggest the authors replace N. ceranae with Nosema ceranae since it is a subtopic

Replaced V. ceranae with Vairimorpha ceranae.

Page 22, Line 343-344: “Effects of different diets and N. ceranae infection on the gut bacterial

microbiota ”

Please replace N. ceranae with Nosema ceranae

Replaced V. ceranae with Vairimorpha ceranae.

Figure 3: Please provide a better solution for the graphs and figure legend of all statistics

Improved

Figures 5 and 6: Please provide the different letters showing the statistical difference in the data in Figure 5 and 6 instead of * or **

Now we use different letters showing the statistical difference in the data in Figure 5 and 6, respectively.

Discussion: I suggest that the authors improve the discussion part by making it shorter, emphasizing, and clarifying the scientific contribution of this paper.

Following the reviewer’s suggestions, we have shortened the discussion, focusing and clarifying the contribution of our work.

References: Please take care of scientific names, make sure they are in italic form.

Supporting information: Please take care of scientific names, make sure they are in italic form.

We used Zotero to manage the references in our manuscript. All citations were manually reviewed and adjusted as needed. We also verified the reviewer’s suggested references throughout our the Supporting Information files.

Reviewer #2:

1. The Nosema ceranae is now considered as Vairimorpha ceranae, therefore, the author should correct this genus everywhere in this manuscript.

The reviewer is right, revising recently published taxonomic papers and the databases like GenBank, Mycobank, and Index Fungorum use Vairimorpha as valid genus and not Nosema. Therefore, we corrected the genus name across the complete manuscript.

2. The major concern of this manuscript is the author mentioned the establishment of N. ceranae infection appeared limited, likely due to low spore viability. In my view point, the microsporidia infection in honey bee should be the basic experiment of this study, which the author should overcome this problem before they trying to publish this paper, otherwise the V. ceranae infection test should be removed in this manuscript and focus on the Ganoderma australe extract on gut microbiota and immune gene expression in honey bees.

We appreciate the reviewer’s valuable comment. As indicated, the establishment of Vairimorpha ceranae infection appeared limited, likely due to low spore viability. However, since the funding source required us to obtain experimental results during the spring–summer season of the project, we were unable to conduct preliminary infection trials with honey bees prior to initiating the assays. In addition, we did not have sufficient resources to acquire intestinal epithelial cell cultures of honey bees for in vitro assays to characterize the infective capacity of V. ceranae spores.

Although the infection level was lower than expected (despite the inoculum being obtained from colonies heavily affected by nosemosis), the presence of the causal agent of the disease still produced measurable effects on the immune system and bacteriome compared with non-infected honey bees. Therefore, we consider that the infection component remains relevant to the interpretation of the physiological responses observed in this study.

3. Table 1: The standard V. ceranae infection process should observe more than 2-weeks (14~21days) and therefore can see the pathogenesis of V. ceranae.

Previous studies investigating the honey bee immune system, such as Glacinic et al. (2017), quantified gene expression levels up to 12 days post-infection, while Aufauvre et al. (2014) collected samples at 1 and 7 days after treatment application. Both studies were published in PLOS ONE. The following Table summarizes some additional publications and the corresponding observation periods used in each study:

Reference Journal Summary

Antúnez et al. (2009)

“Immune suppression in the honey bee (Apis mellifera) following infection by N. ceranae” Environmental Microbiology The RNA expression levels of seven genes were determined 4 and 7 days after N. ceranae infection.

Antúnez et al. (2013)

“Differential expression of vitellogenin in honey bees (Apis mellifera) with different degrees of N. ceranae infection” Journal of Apicultural Research Newly emerged bees from both colonies were artificially infected with N. ceranae and seven days after infection

---

## [Decision Letter · Decision Letter 1]

19 Jan 2026

Exploratory study on the impact of Ganoderma australe extract on gut microbiota and immune gene expression in honey bees exposed to Vairimorpha ceranae

PLOS One

Dear Dr. Garnica,

Thank you for submitting your manuscript to PLOS ONE. After careful consideration, we feel that it has merit but does not fully meet PLOS ONE’s publication criteria as it currently stands. Therefore, we invite you to submit a revised version of the manuscript that addresses the points raised during the review process.

We look forward to receiving your revised manuscript.

Kind regards,

Kai Wang

Academic Editor

PLOS One

Journal Requirements:

Reviewer's Responses to Questions

**Comments to the Author**

Reviewer #1: (No Response)

2. Is the manuscript technically sound, and do the data support the conclusions?

Reviewer #1: Yes

3. Has the statistical analysis been performed appropriately and rigorously?

Reviewer #1: Yes

4. Have the authors made all data underlying the findings in their manuscript fully available?

Reviewer #1: Yes

5. Is the manuscript presented in an intelligible fashion and written in standard English?

Reviewer #1: Yes

Reviewer #1: After I looked through the manuscript titled “Exploratory study on the impact of Ganoderma australe extract on gut microbiota and immune gene expression in honey bees exposed to Vairimorpha ceranae”. The manuscript looks good to me; however, the manuscript needs to be revised as a minor revision. Please see the comments below.

Abstract:

Looks good to me; however, I suggest the authors replace honeybee with honey bee throughout the manuscript.

Comment 1: Line 8-9 “…..honey bees subjected to experimental exposure to V. ceranae 1 x 106 spores/ml.” I suggest the author provide the spore load (number of spores per bee) instead of the number of spores per ml.

Introduction:

Overall, the introduction looks good to me.

Materials and methods:

The authors used the standard methods and showed scientific rigor. However, some minor concerns as follows.

Comment2: Page 18, lines 102-115: Fruiting bodies collection and pure cultures

Please provide the conditions, such as temperature, humidity, and period for the culture.

Comment 3: Page 18, line 124: Molecular identification of fungal isolates.

Please use the xg instead of “at 3450 rpm”

Comment 4: Page 20, lines 140-156: Fungal extract

Please provide the internal standard for HPLC that the authors used for analysis.

Comment5: line 210: Five bees of each cage were fed individually (10 µl), and lines 212-213, then further fed by group feeding? (2 mL per cage?). Please provide when group feeding was provided.

Results:

Please provide a better resolution of all figures.

Discussion:

Looks pretty good and make senses to me.

**Do you want your identity to be public for this peer review?** For information about this choice, including consent withdrawal, please see our Privacy Policy

Reviewer #1: No

---

## [Author Response · Author response to Decision Letter 2]

9 Feb 2026

Reviewer #1: After I looked through the manuscript titled “Exploratory study on the impact of Ganoderma australe extract on gut microbiota and immune gene expression in honey bees exposed to Vairimorpha ceranae”. The manuscript looks good to me; however, the manuscript needs to be revised as a minor revision. Please see the comments below.

Abstract:

Looks good to me; however, I suggest the authors replace honeybee with honey bee throughout the manuscript.

Changed throughout the manuscript.

Comment 1: Line 8-9 “…..honey bees subjected to experimental exposure to V. ceranae 1 x 106 spores/ml.” I suggest the author provide the spore load (number of spores per bee) instead of the number of spores per ml.

1 × 10⁴ spores per bee. This load was administered by individual feeding.

Introduction:

Overall, the introduction looks good to me.

Fine

Materials and methods:

The authors used the standard methods and showed scientific rigor. However, some minor concerns as follows.

Comment2: Page 18, lines 102-115: Fruiting bodies collection and pure cultures

Please provide the conditions, such as temperature, humidity, and period for the culture.

We provided now the culture conditions, including temperature, humidity, and duration; in our study, cultures were maintained at 23°C and 60% humidity for a maximum of 30 days.

Comment 3: Page 18, line 124: Molecular identification of fungal isolates.

Please use the xg instead of “at 3450 rpm”

We have now changed the unit to ×g, which in our study corresponds to a centrifugal force of 253 × g.

Comment 4: Page 20, lines 140-156: Fungal extract

Please provide the internal standard for HPLC that the authors used for analysis.

A ganoderic acid A standard (CAS No. 81907-62-2) was used at a concentration of 50 µg/mL and dissolved in 100% acetonitrile. The standard was purchased from Sigma-Aldrich (Chile).

Comment 5: line 210: Five bees of each cage were fed individually (10 µl), and lines 212-213, then further fed by group feeding? (2 mL per cage?). Please provide when group feeding was provided.

Improved

Results:

Please provide a better resolution of all figures.

All figures were newly generated to ensure high resolution suitable for publication in PLOS ONE.

Discussion:

Looks pretty good and make senses to me.

7. PLOS authors have the option to publish the peer review history of their article (what does this mean?). If published, this will include your full peer review and any attached files.

Do you want your identity to be public for this peer review? For information about this choice, including consent withdrawal, please see our Privacy Policy.

Reviewer #1: No

---

## [Editor Report · Decision Letter 2]

17 Feb 2026

Exploratory study on the impact of Ganoderma australe extract on gut microbiota and immune gene expression in honey bees exposed to Vairimorpha ceranae

PONE-D-25-39736R2

Dear Dr. Garnica,

We’re pleased to inform you that your manuscript has been judged scientifically suitable for publication and will be formally accepted for publication once it meets all outstanding technical requirements.

Kind regards,

Kai Wang

Academic Editor

PLOS One
---

## [Editor Report · Acceptance letter]

PONE-D-25-39736R2

PLOS One

Dear Dr. Garnica,

I'm pleased to inform you that your manuscript has been deemed suitable for publication in PLOS One. Congratulations! Your manuscript is now being handed over to our production team.

Kind regards,

on behalf of

Dr. Kai Wang

Academic Editor

PLOS One